# Climate Change and Psychiatry: The Correlation between the Mean Monthly Temperature and Admissions to an Acute Inpatient Unit

**DOI:** 10.3390/ijerph21070826

**Published:** 2024-06-25

**Authors:** Nicola Rizzo Pesci, Elena Teobaldi, Giuseppe Maina, Gianluca Rosso

**Affiliations:** 1Department of Neurosciences “Rita Levi Montalcini”, University of Turin, 10126 Turin, Italy; nicola.rizzopesci@unito.it (N.R.P.); elena.teobaldi@unito.it (E.T.); giuseppe.maina@unito.it (G.M.); 2Psychiatric Unit, San Luigi Gonzaga University Hospital, 10043 Turin, Italy

**Keywords:** global warming, temperature, mental health, major depressive disorder, bipolar disorder, schizophrenia

## Abstract

Background: Psychiatric disorders are large contributors to the global disease burden, but research on the impact of climate change on them is limited. Our aim is to investigate the correlation between temperature and exacerbations of psychiatric disorders to help inform clinical management and future public health policies. Methods: Temperature records for the summer months from 2013 to 2022 were obtained from the meteorological station of the Department of Physics of Turin University. Data on patients admitted to the acute psychiatric unit were extracted from registries of San Luigi Gonzaga University Hospital (Turin, Italy). Regression analyses were used to investigate the correlation between temperature and number of admissions and to test for confounding variables. Results: A total of 1600 admissions were recorded. The monthly temperature and number of admissions were directly correlated (*p* = 0.0020). The correlation was significant for the subgroup of admissions due to Bipolar Disorders (*p* = 0.0011), but not for schizophrenia or major depressive disorder. After multiple regression analyses, the effect of temperature remained significant (*p* = 0.0406). Conclusions: These results confirm the impact of meteorological factors on mental disorders, particularly on BD. This can contribute to personalised follow-up and efficient resource allocation and poses grounds for studies into etiopathological mechanisms and therapeutic implications.

## 1. Introduction

In recent years, the global climate has faced drastic changes, partly induced by human activities. New peaks of global temperature have been reached in recent years, and scientists predict a progressive and unavoidable worsening in the years to come [1]. Last year was the warmest solar year on record [2], and the 6 July 2023 was the warmest day ever recorded, with a global average of 17.08 °C [3]. Climate change is making extreme climatic events such as heat waves, floods, and draughts more common [4,5] and is altering seasonal patterns [6].

The impact of temperature and hence of climate change on morbidity has been studied for over a decade, with a clear focus on cardiovascular disease, endocrine and metabolic disorders, and other conditions related to physical health and infective disorders [7]. Seasonal patterns have been observed in the gut microbiota of patients living with inflammatory bowel diseases [8], as well as in symptoms of multiple sclerosis [9], and temperature extremes have been associated with mortality due to respiratory disorders [10]. Many authors have specifically highlighted the impact of climate changes on morbidity. Available data suggest that rising temperatures lead to increased cardiovascular mortality through increased blood pressure and viscosity, increased heart rate, and endothelial damage [11]. Extreme weather events have been shown to increase morbidity and mortality in patients living with diabetes [12]. The impact of pollution, greenhouse gases, and climatic changes on reproductive health has also been explored, with reports showing damaging effects of climate change on fertility and pregnancy outcomes [13].

Mental disorders are among the leading causes of the disease burden worldwide, and their impact in terms of prevalence and disability has been increasing in the last 30 years [14], but little attention has been paid to the effects of climate change on mental health. Psychiatric conditions can lead to suicidal ideation and self-harm [15], as well as psychomotor agitation [16], which often lead to presentation to emergency departments and admission, whose rates are also increasing [17].

There is a tight interconnection between mental health and the climate, specifically in terms of seasonal cycles, light exposure, and temperature. Many authors have reported seasonal patterns in the course of psychiatric disorders, with rises in admissions, involuntary admissions, and suicidal ideation in spring/summer [18]. Some reports have also suggested that different diagnostic groups face peaks in admissions at different times of the year, with exacerbations of Bipolar Disorder (BD) in summer, schizophrenia (SCZ) in winter, and major depressive disorder (MDD) in early winter [19,20]. Nevertheless, available data support the notion that seasonal patterns are particularly characteristic of bipolar spectrum disorders [21]. Also, light–dark cycles are of high relevance for mental health conditions, especially for mood disorders. Data suggest that the photoperiod and sunlight exposure are directly correlated with the rate of admission for manic or hypomanic episodes [22,23]. Daylight exposure also influences sleep patterns, which play a crucial role in mental disorders. Sleep disruption is a symptom of many psychiatric conditions, and the modulation of light exposure and sleep has therapeutic applications [24]. Bright light therapy and sleep deprivation therapy have, indeed, proven efficacy in seasonal affective disorder [25], as well as in unipolar and bipolar depression, and act through the normalisation of sleep patterns, as well as a plethora of biological mechanisms at the cellular level [26,27,28]. Research concerning the relationship between the temperature and mental health also exists, but the results are not univocal. Few papers have described direct correlations of maximum and mean ambient temperatures with emergency presentations for psychiatric disorders [23,29], with involuntary admissions [30] and even with mortality in psychiatric patients [31]. On the other hand, some authors found small or non-significant impacts of meteorological patterns on admission rates [32,33,34]. In a recent review, the authors gathered data on the association between admissions for schizophrenia and the temperature. Despite most of the included studies reporting an association between higher temperatures and admission rates for schizophrenia, studies are heterogeneous and often contradicting, and further research on the matter has been suggested [35].

Despite the burden of psychiatric disorders, their known interconnection with environmental and climatic factors, and gaps in the understanding of the mechanisms underlying such interconnection, research on the impact of climatic changes on mental health is scarce, and its generalisability is hindered by geographical factors. Air pollution, high temperatures, draughts, and extreme precipitation events have been associated with an increased suicide risk [36], and one large American study based on nearly 2 million observations suggested an impact of multi-year rises in temperatures on mental health issues. Nevertheless, the results of the latter study were drawn from data that are more than 10 years old now, and the authors could not differentiate acute from chronic mental issues, nor could they identify the impact of climatic changes on different diagnostic groups [37]. In 2018, a systematic review identified 35 studies assessing the impact of temperature on mental health outcomes, but only 2 were conducted in Mediterranean Europe, and none investigated the effects of the rise in temperature observed in recent decades [38]. Therefore, further research efforts are necessary to confirm the putative impact of global warming on mental health.

The aim of this study is to contribute to the body of research concerning the effects of climate change on psychiatric disorders. With this in mind, we aimed to test the hypothesis that the ambient temperature would be correlated with exacerbations of psychiatric disorders. We intended to achieve this by analysing the correlation between the mean monthly temperatures and admissions to our acute psychiatric inpatient unit during the summer months over a 10-year period. Clarifying the correlation between temperature and mental health exacerbations could foster improvements in clinical management and resource allocation both at a community service level and from a public health policy point of view. 

## 2. Materials and Methods

### 2.1. Study Design

The present study is a descriptive, cross-sectional study. Data on mean monthly temperatures and on admissions to our clinic were collected retrospectively for the summer months of a 10 year period. The study included data from all patients admitted during the observation period to the acute psychiatric unit of San Luigi Gonzaga University Hospital in Orbassano, Turin, Italy. The total observation period was 40 months, i.e., the months from June to September of the years from 2013 to 2022. The data collection and analyses for the present study were conducted with the approval of the local Ethics Committee (Comitato Etico Interaziendale San Luigi Gonzaga; approval code: 7119/Tit:02/Cat:06).

### 2.2. Data Extraction

Temperature: Data on daily temperatures for the observation period were downloaded from online registries of measurements from the meteorological station of the Department of Physics of the University of Turin. Data were probed for outlying or missing values and were found to be consistent and complete. Mean monthly temperatures were computed for statistical analyses.

Admissions: Data regarding socio-demographic (i.e., age and gender) and clinical (i.e., main diagnosis at discharge, date, duration, and compulsoriness of admission) characteristics were extracted from hospital registries. These were only obtained for patients who had given consent for observational research based on stored data, and they were fully anonymised at the moment of extraction by removing identifiable information and assigning a study number to each admission. Admissions for which the socio-demographic or clinical characteristics were not recorded were excluded from the analyses. Diagnoses at discharge in our hospital registry are encoded according to the ICD-9 classification. For the present study, diagnoses were divided into four mutually exclusive groups: schizophrenia and related disorders; Bipolar and related disorders; major depressive disorder; and other. “Other” diagnoses included personality disorders, obsessive compulsive disorder, substance use disorder, eating disorders, and neurocognitive disorders. Admission data were grouped according to month and year (*N* = 40) and diagnostic group, and the number of admissions for each diagnostic group and for each month were computed for the analyses.

### 2.3. Statistical Analyses

All statistical analyses were carried out on GraphPad Prism 8 (GraphPad, Boston, MA, USA). The value of statistical significance was set at *p* < 0.05. Power analysis was performed to determine the appropriateness of the sample size. This yielded a power of 0.87 for *N* = 40 and an estimated effect size R^2^ = 0.2. A Shapiro–Wilk test was used to test the normality of distribution for continuous variables. An independent T test was used to compare the means of independent measures between two groups. One-way ANOVA was used when comparing means between more than two groups. For the analysis of correlation between two continuous variables (e.g., temperature and number of admissions), we performed linear regression analyses and computed Pearson correlation coefficients (R^2^). A multiple linear regression model was used to assess the impact of potential confounding variables.

## 3. Results

### 3.1. Sample Characteristics and Descriptive Statistics

A total of 1600 admissions to our psychiatric ward were recorded over the observation period. The mean number of admissions per month was 40 (±9.89). The mean (±Standard Deviation (SD)) age of the sample was 45.74 (±15.51) years; 722 of the patients (45.13%) were females; the mean duration of hospitalisation was 9.99 (±8.95) days; and the rate of involuntary admission was 7.38% (*N* = 118).

Concerning the main diagnosis, 556 (34.75%) patients had Bipolar and related disorders, 325 (20.31%) had schizophrenia and related disorders, 231 (14.44%) had major depressive disorders, and 488 (30.50%) had other diagnoses (Table 1). The mean temperature throughout the observation period was 22.24 °C (±2.34). The lowest mean monthly temperature recorded was 17.70 °C (September 2015), and 28 admissions were recorded in that month. The warmest month was also the one with the highest number of admissions. This was July 2022, with 27.10 °C and 64 admissions. Figure 1 shows the trend in mean monthly temperature (left Y axis) and total number of admissions for each month (right Y axis) (Figure 1).

### 3.2. Total Number of Admissions According to Month

The mean numbers of admissions for each month of any year are reported in Figure 2. No significant difference was observed between the mean numbers of admissions for each month (*p* = 0.3494) (Figure 2).

### 3.3. Temperature and Number of Admissions

We performed a linear regression analysis, setting the mean monthly temperature as the independent variable and monthly number of admissions as the dependent variable. The regression coefficient was 1.998 (95% Confidence Interval (CI) 0.7785 to 3.218, *p* = 0.0020), and the Pearson correlation coefficient (R^2^) was 0.2245 (Figure 3).

### 3.4. Temperature and Number of Admissions According to Diagnosis

Linear regression analysis was performed between the mean monthly temperature and number of admissions according to each diagnostic group. The regression coefficient was 0.1231 (95% CI −0.3576 to 0.6039, *p* = 0.6070) for admissions of patients with schizophrenia and related disorders, 1.410 (95% CI 0.6052 to 2.215, *p* = 0.0011, R^2^ = 0.2487) for bipolar and related disorders, and 0.3033 (95% CI −0.01183 to 0.6184, *p* = 0.0588) for major depressive disorder (Figure 4).

### 3.5. Temperature and Involuntary Admissions

No significant correlation was found when performing a linear regression analysis between the temperature and monthly number of involuntary admissions (Y = −0.01314 × X + 3.242; *p* = 0.9284; R^2^ = 0.0002151) (Figure 5A).

### 3.6. Temperature and Length of Stay

No significant correlation was found when performing a linear regression analysis between the mean monthly temperature and length of hospitalisation (Y = −0.06038 × X + 11.62; *p* = 0.6739; R^2^ = 0.004709) (Figure 5B).

### 3.7. Multiple Regression Model

Multiple regression analysis was performed to test for potential confounding effects of month, year, mean duration of admission, and percentage of female patients. The model yielded a significant correlation between year and number of admissions (*p* = 0.0008). The correlation between temperature and number of admissions remained significant (*p* = 0.0406). The overall R^2^ of the model was 0.6756 (Table 2).

## 4. Discussion

In recent years, climate change has become increasingly evident. Extreme climatic events such as heavy precipitations, floods, prolonged draughts, and heat waves are becoming more common [4], and global temperatures are rising [2]. The impact of rising temperatures and adverse meteorological events on health is also increasingly considered [7,39]. Despite the available knowledge on the interconnection between mental health and climatic conditions [20,32,40], few authors have investigated the impact of climatic changes on psychiatric disorders. Indeed, our comprehension of this impact is hampered by a paucity of data, contradictory results, and poor generalisability due to the limited geographical diffusion of these investigations [38].

To our knowledge, this is the first study explicitly examining the impact of rising temperatures on mental health outcomes in Mediterranean Europe throughout a long period of observation. A recent systematic review of 35 studies [38] identified 2 previous analyses of the correlation between temperature and admissions in Spain, but they either focused on admissions for dementia [41], or they considered a short time span [42]. In both cases, analyses were conducted on data that are now older than 15 years, and the rising trend in temperatures was not taken into consideration.

The present study investigated the correlation between the mean monthly temperatures in the summer months and admissions to our acute psychiatric unit in Turin, Italy. The highest mean monthly temperature recorded throughout the observation period was in July 2022 at 27.10 °C. This is more than 2 SD higher than the mean temperature observed throughout this study (22.24 °C ± 2.34), and previous reports have noted a prominent hazard and increased all-cause mortality in Italy when temperatures between the 90th and 99th centile have been reached [43]. Interestingly, July 2022 was also the month with the highest number of admissions being observed. Our data show an oscillating but overall increasing trend in mean summer temperatures in Turin throughout the observation period, paralleled by the trend in admissions to our inpatient unit. A relevant background consideration is that this suggests that the variations in the number of admissions are not merely due to a secular drift, which might be explained by confounding factors such as an overall increased prevalence of psychiatric conditions or increased efficiency in inpatient turnover. Moreover, the similarity in the evolution of summer temperatures and summer admissions, together with the non-significant difference in the total number of admissions for each month of any year, suggests that the observed correlations are not only explained by the potential confounding effect of seasonal patterns of psychiatric disorders. This was tested by applying a multiple regression model that included the month as an independent variable, which showed no significant correlation between the month and number of admissions. Our analyses yielded a significant correlation between the mean monthly temperature and the number of admissions to our psychiatric inpatient unit. The R^2^ value suggests that only a small percentage of the variance in the number of admissions in our sample is explained by changes in monthly temperatures. This is expected given the complex nature of mental health disorders and the vast array of potentially contributing factors, of which temperature might be one. This would highlight the importance of considering mental health outcomes when discussing the adverse effects of global warming and is in line with previous reports [29]. Contrary to what Jahan and colleagues reported [35], we found no association between the temperature and admissions when looking into the sample of patients with SCZ. It has to be noted that Jahan and colleagues stressed the heterogeneity of results on the matter and suggested further research themselves in order to confirm or dismiss the notion that the course of SCZ is influenced by the temperature. Moreover, more recent studies found either no [19] or very little [33] impact of high temperatures on psychotic disorders, and our study might not be powered enough to detect such a small effect. On the other hand, our results confirmed the impact of the temperature on the course of BD, as suggested by other pieces of research [23], but not of MDD. The correlation observed for BD and not for MDD is in accordance with the knowledge of a more prominent influence of seasonal patterns and meteorological factors on bipolar spectrum- compared with unipolar affective disorders [20,21]. Seasonal peaks in depressive episodes are commonly reported but tend to occur in winter seasons [19] and, therefore, are likely to be less influenced by high ambient temperatures. A multiple regression analysis was performed to control for the impact of other independent variables. This included month of the year, year, mean duration of admission, and percentage of female patients admitted. The model yielded a significant correlation between the year and number of admissions and confirmed the correlation between the temperature and number of admissions.

In our opinion, the implications of the presented results stretch beyond epidemiological considerations. The relationship between increasing temperatures and rising numbers of admissions might shed a light on pathophysiological mechanisms contributing to the onset or exacerbation of major psychiatric disorders and in particular BD. Exposure to high temperatures can induce accumulation of reactive oxygen species and alterations in blood–brain barrier permeability [39]. This is especially relevant given the available research on the role of inflammation [44], oxidative stress [45], and blood–brain barrier leakiness [46] in BD’s etiopathogenesis and course. Moreover, increased temperature and perspiration can alter lithium pharmacokinetics [47] and potentially their therapeutic effect.

Understanding the relationship between the temperature and exacerbations of BD can also inform patient monitoring. The disease course of BD is influenced by a number of circadian cycles such as melatonin peaks, cortisol levels, body temperature, sleep–wake alternation, and chronotype [40]. It is undoubtedly relevant to know a patient’s sleep habits and to consider their exposure to melatonin or cortisol in order to adequately approach a clinical picture. Similarly, if the negative impact of rising temperatures on BD was confirmed, this should be taken into account when managing an exacerbation of the disorder. Knowledge of underlying mechanisms can allow for the correct interpretation of patient’s history, and most of all, it can facilitate accurate follow-up. As public health resources are lacking in many areas of the globe, and those assigned to mental health are consistently inferior to demand [48], finding ways to make resource allocation more efficient is becoming increasingly crucial. The possibility of reasonably predicting which patients are more likely to experience a disease recrudescence depending on the season and temperature trend can, therefore, become a valuable asset for psychiatrists.

Future studies should therefore look into the potential mechanisms underlying this interaction and assess how treatment and management strategies can modulate this relationship. This will be of paramount importance to effectively address the challenges that climate change poses to mental health patients and services worldwide.

The presented results and the discussed implications have to be understood in light of the limitations of our study. The observational cross-sectional design of our study limits the inference of causal relationships between the exposure (i.e., higher temperatures) and the putative effect (i.e., admissions). Nevertheless, studies on weather variables, especially those considering historical measurements over long periods of time, can only be observational in design. While there is a need for prospective studies on this subject, we judged that the analysis and dissemination of available cross-sectional data would be crucial to provide, without delay, exploratory insights that can foster appropriate hypothesis formulation and design for future studies. Our observation period was limited to the summer months, while taking into consideration the fact that the whole year might be more explicative of the true effect of temperature on psychiatric presentations. We decided to only consider the summer months in order to minimise the potential confounding effect of seasonality and because the effects of global warming are more evident in the summer months [2]. Psychiatric disorders in general, and BD in particular, are known to show seasonal patterns [18,20,35], but in order to probe the effect of temperature changes, we decided to focus on a single season. We opted for a monthly resolution of our measurements, which limited our data points to 40. While narrower time windows (e.g., weekly or daily) would increase the number of data points, improve the significance, and potentially provide more information, a monthly time frame allows us to observe the effects of the exposure, even if lagged in time. Given the limited availability of reports on this subject and the lag expected in complex conditions such as psychiatric disorders, monthly measurements were preferred in order to detect correlations that would need to be looked into in further detail and with improved resolution in future studies. Moreover, the limited amount of information that we could extract for this study only allowed us to control for a few variables. We could not control for other potentially relevant confounding factors such as other clinical and socio-demographical patient-level information (ongoing treatment, comorbidity, age, occupation, relationship status), nor for other variables such as the total number of admissions in our hospital or region, health policies, and the overall prevalence of mental health disorders in our area. Future analyses should investigate the role of these variables as well. Our data on admissions suffer from a ceiling effect imposed by the maximum number of beds in our inpatient unit. The number of presentations to the emergency departments for psychiatric complaints would represent a more sensible parameter and would work around the ceiling bias. On the other hand, admissions were preferred for this study in order to increase the specificity for severe exacerbations of psychiatric disorders, i.e., those that require admission. The presented results have an intrinsic limit in their external validity due to geographical reasons. They are obtained from a single centre and can be a relevant and novel contribution to the understanding of the described phenomena in the Mediterranean area. Nevertheless, they must be interpreted in the context of other reports from diverse parts of the world, especially when pondering their etiopathological and therapeutical implications.

## 5. Conclusions

This paper describes a direct correlation between the mean monthly temperature in summer and the number of admissions to an acute psychiatric inpatient unit in Italy. If corroborated by other studies, this notion would be of the utmost relevance in view of the climate changes that we are facing. Clarifying the effect of rising temperatures on mental health could contribute to further understanding of pathophysiological mechanisms, to the development of more effective relapse prevention strategies at a service level, and to dynamic resource allocation from a public health point of view.

## Figures and Tables

**Figure 1 ijerph-21-00826-f001:**
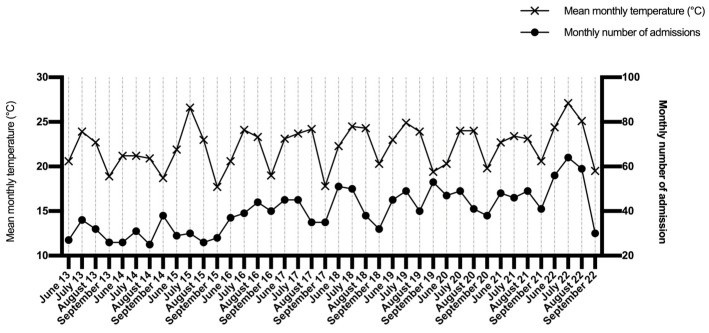
Trends in mean temperatures and total number of admissions per month.

**Figure 2 ijerph-21-00826-f002:**
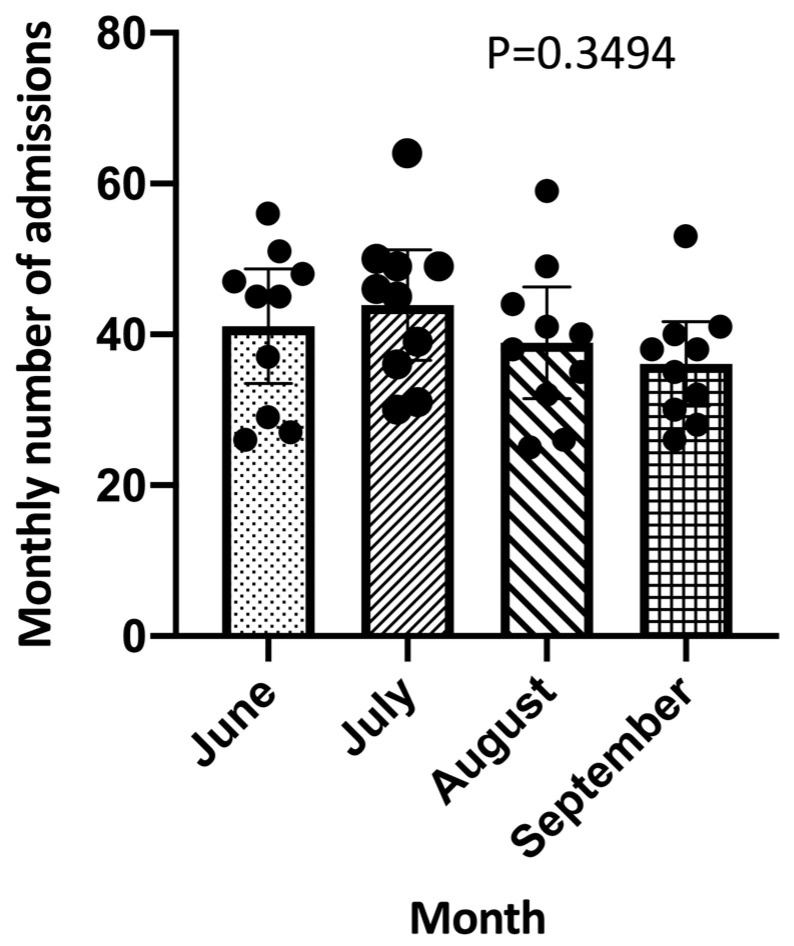
Comparison of total number of admissions for each month of any year.

**Figure 3 ijerph-21-00826-f003:**
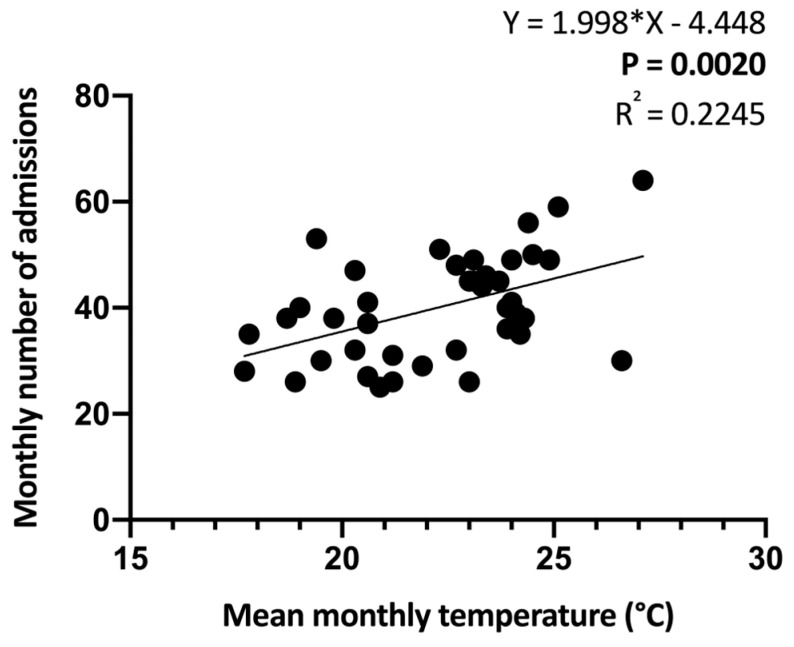
Linear regression analysis between mean monthly temperature and monthly number of admissions.

**Figure 4 ijerph-21-00826-f004:**
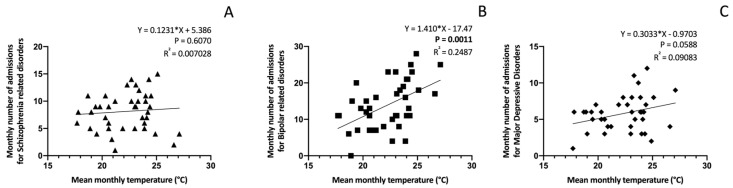
Linear regression analyses between mean monthly temperature and monthly number of admissions for schizophrenia (**A**), Bipolar Disorder (**B**), and major depressive disorder (**C**).

**Figure 5 ijerph-21-00826-f005:**
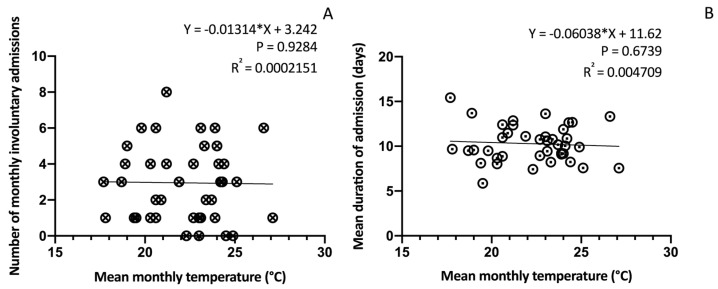
Linear regression analysis between mean monthly temperature and monthly number of involuntary admissions (**A**) and mean duration of admission (**B**).

**Table 1 ijerph-21-00826-t001:** Socio-demographic and clinical characteristics of the sample.

Total number of admissions	1600
Age, mean (+/− SD)	45.74 (+/− 15.51) years
Gender, *n* (%)	
Male	878 (54.87)
Female	722 (45.13)
Main diagnosis, *n* (%)	
BD	556 (34.75)
MDD	231 (14.44)
SCZ	325 (20.31)
Type of admission, *n* (%)	
Voluntary	1482 (92.62)
Involuntary	118 (7.38)
Duration of admission, mean (+/− SD)	9.99 (+/− 8.95) days

**Table 2 ijerph-21-00826-t002:** Multiple regression model.

Independent Variable	Slope	95% CI	*p*-Value
Intercept			
Mean temperature	1.051	0.04758 to 2.055	0.0406
Month	−0.9861	−2.986 to 1.014	0.3235
Year	1.840	0.8202 to 2.861	0.0008
Mean duration of admission	−1.173	−2.458 to 0.1119	0.0722
% of female patients	−0.07156	−0.3187 to 0.1756	0.5602
R^2^ = 0.6756			
F (5, 34) = 14.16			
*p* < 0.0001			

## Data Availability

Data supporting the findings of the present study are available upon request.

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
