# Peer review of "Climate Change and Psychiatry: The Correlation between the Mean Monthly Temperature and Admissions to an Acute Inpatient Unit"

_ijerph, 2024, doi:10.3390/ijerph21070826_

Round 1

Reviewer 1 Report

Comments and Suggestions for Authors

This paper is well- written really but  I suggest the following.
Conceptual
"This study intends to investigate the correlation between rising temperatures and admissions for acute exacerbations of psychiatric diseases."
Clearly state the reasons this association is significant and how it might affect public health or clinical practice.
"This study aims to investigate the correlation between rising temperatures and admissions for acute exacerbations of psychiatric disorders, which could inform clinical strategies for managing mental health in the context of climate change," says one researcher.

Methodology : Add specifics on the sample size, data management, and any controls or adjustments applied in the regression analysis.

For instance, "We gathered information for summer months 2013 through 2022 including mean monthly temperatures from the University of Turin's meteorological station and admission records from the acute psychiatric ward at San Luigi Gonzaga University Hospital. Using linear regression, 1600 admissions in all were examined under control for possible confounders like comorbidities and socioeconomic levels.

3. Discuss the strength of the correlations—that is, the correlation coefficients—and offer more background regarding the non-significant results.
4.  Talk about more general consequences and possible future directions of research in clinical practice.

Future studies, for instance, should look at fundamental causes and assess treatments to reduce mental health concerns associated to climate change."

Design of the study.
5. Explain your choice of a descriptive, cross-sectional design and go over its restrictions. Think about giving more background regarding the demographic and research environment.
6. Describe more precisely how temperature data were managed and evaluated in the data extracting process. Discuss any actions done to guarantee the accuracy of these records and the justification behind selecting the summer months.
7. Admission data handling: Describe the anonymizing technique in more particular terms and how data consistency was kept. Add any requirements for either inclusion or exclusion of patient records.
8. Discussion: Talk about all limits holistically, including generalizability, possible biases, and sample size.

Comments on the Quality of English Language

The quality of the English language is okay be can be improved. Minor editing here and there. Ensure consistent formatting throughout the paper and correct any typographical errors.

Author Response

Thank you very much for your review and for your precious feedback. We made our best to improve the methods, results and conclusion sections according to your suggestions, which we think improved our paper’s clarity, overall structure, and correct interpretation of the presented results. Please find point-by-point answers in the attached file.

Reviewer 2 Report

Comments and Suggestions for Authors

This study is focused on the effect of climate change, specifically, temperature on the hospital admission rate for mental health disorders. Authors only focused on summer months to avoid confounding effect of seasonal variation. However, even within summer months, only monthly means of temperature and hospital admission. Analysis of daily parameters would have provided better understanding of the effect of temperature.

Author Response

Thank you very much for reviewing our paper and for your valuable feedback. We made our best to improve the methods and conclusion sections according to your suggestions, and to clarify in more details the choice of monthly temperatures while also describing more thoroughly its implications in the limitation section. We think that the suggested amendments improved our paper’s overall clarity, structure, and proper interpretation. We provided a more detailed explanation of the reasons behind the choice of summer months only in the limitation section (“We opted for a monthly resolution of our measurements, which limited our data points to 40. While narrower time windows (e.g. weekly or daily) would increase data points, improve significance, and potentially provide more information, a monthly time-frame allows to observe the effects of the exposure even if lagged in time. Given the limited availability of reports on this subject and the lag expected in complex conditions such as psychiatric disorders, monthly measurements were preferred in order to detect correlations that would need to be looked into with further detail and with improved resolution in future studies.”)

Reviewer 3 Report

Comments and Suggestions for Authors

Author Response

Thank you very much for reviewing our paper and for your valuable feedback. We made our best to improve the methods, results and conclusion sections according to your suggestions, which we think improved our paper’s clarity, overall structure, and correct interpretation of the presented results. Please see the attachment for point-by-point responses.
